# Lipid Transporters Beam Signals from Cell Membranes

**DOI:** 10.3390/membranes11080562

**Published:** 2021-07-26

**Authors:** Miliça Ristovski, Danny Farhat, Shelly Ellaine M. Bancud, Jyh-Yeuan Lee

**Affiliations:** 1Department of Biochemistry, Microbiology and Immunology, Faculty of Medicine, University of Ottawa, Ottawa, ON K1H 8M5, Canada; mrist104@uottawa.ca (M.R.); dfarh021@uottawa.ca (D.F.); sbanc085@uottawa.ca (S.E.M.B.); 2Translational and Molecular Medicine Program, Faculty of Medicine, University of Ottawa, Ottawa, ON K1H 8M5, Canada; 3Biomedical Sciences Program, Faculty of Science, University of Ottawa, Ottawa, ON K1H 6N5, Canada

**Keywords:** P4-ATPase, ABC transporter, phospholipid, cholesterol, membranes, cellular signaling

## Abstract

Lipid composition in cellular membranes plays an important role in maintaining the structural integrity of cells and in regulating cellular signaling that controls functions of both membrane-anchored and cytoplasmic proteins. ATP-dependent ABC and P4-ATPase lipid transporters, two integral membrane proteins, are known to contribute to lipid translocation across the lipid bilayers on the cellular membranes. In this review, we will highlight current knowledge about the role of cholesterol and phospholipids of cellular membranes in regulating cell signaling and how lipid transporters participate this process.

## 1. Introduction

The living plasma membrane (PM) is a fluid lipid bilayer, forming a structural barrier between the cytoplasm and the environment [1]. The lipids interact with proteins through hydrophobic and Coulomb forces, allowing membranes to create different domains based on the lipid type components. The domains, such as lipid rafts and membrane curvature formation, then conform to different structures with specific functions [2,3,4]. In addition, the PM is highly dynamic and actively maintains a trans-bilayer asymmetry, wherein its constituents are continuously and selectively synthesized, transferred, and trafficked to the membrane [5]. In particular, the outer leaflet is mostly composed of cholesterol, phosphatidylcholine (PC), and sphingomyelin (SM), whereas the inner leaflet is enriched with phosphatidylethanolamine (PE), phosphatidylserine (PS), and phosphatidylinositol (PI) [6,7,8,9]. The asymmetry in both lateral and in between leaflets is vital for cellular functioning and signaling, with the outer leaflet more tightly packed [9]. A loss of membrane asymmetry is associated with various diseases, including apoptosis [10,11], bleeding disorders [11,12,13,14,15,16], sepsis [15,17], and cancer [15,18,19,20].

Three types of membrane lipid transporters maintain the PM asymmetry: flippases, floppases, and scramblases [21,22]. Flippases transport lipids from the outer to cytosolic leaflet [21,23,24,25]. Floppases transport lipids from the cytosolic to outer leaflets [26]. Scramblases transport lipids in either direction, mainly to diminish lipid asymmetry by randomizing lipid distributions [27,28]. Defective scramblases which expose PS to the outer leaflet have been linked to bleeding disorders such as Scott syndrome since the exposed PS promotes blood coagulation [29]. Type-IV P-type ATPases (P4-ATPases) are amongst the class of flippases; they harness energy from ATP hydrolysis to transport phospholipid from the extracellular to the cytoplasmic leaflet of the PM [30]. Lipid density changes the response of P4-ATPases [31]. Recent studies have also shown that loss of membrane asymmetry in red blood cells is required for myoblast fusion and platelet activation [9]. Similarly, the ATP binding cassette (ABC) transporters function as either floppases or flippases to catalyze the ATP-dependent transport of cholesterol and other related substrates in between the leaflets [28]. In this article, we will highlight the state-of-knowledge about the role of cholesterol and phospholipids of cellular membrane in regulating cell signaling and how P4-ATPase and ABC sterol transporters participate in this process.

## 2. P4-ATPase Phospholipid Transporters

### 2.1. P4-ATPase Structure

P4-ATPases are found across the kingdoms of archaea, eukaryotes, and prokaryotes [32]. They are similar in domain structure but vary in terms of substrates and the N-termini of the polypeptides [33,34]. Here, we will focus on the human and the budding yeast P4-ATPases [32,35,36].

A P4-ATPase, the alpha (α) subunit, is often paired with a beta (β) subunit, such as CDC50A (Figure 1A,B) [37]. The β subunit is generally composed of two transmembrane helices, where there is evidence showing a shared cholesterol-binding site with the α subunit. [38]. Recent cryo-EM studies have provided a structural glimpse on the overall architecture of P4-ATPases [38,39,40,41]. Each monomeric P4-ATPase consists of ten transmembrane domains, TM1 to TM10 [42], and three cytoplasmic domains, the nucleotide binding (N), the phosphorylation (P), and the actuator (A) domains (Figure 1C).

### 2.2. P4-ATPases and How They Affect Membrane Composition

Traditionally, PC, PS and PE are considered the transport substrates for P4-ATPases, such as yeast Drs2 or human ATP8A [37]. While orthologs to each other, Drs2p and ATP8A have distinct regulatory domains at the C-termini. In yeast Drs2p, the C-terminus has an autoinhibitory effect on ATPase activity; however, the human ATP8A2 mediates a regulation mode where the regulatory domain keeps the N and A domains apart in the E2P state (Figure 1C) [38,43]. Such differential regulation may be attributed to the binding of different β subunits. Recently, glucosylceramide (GlcCer) was discovered to be transported by ATP10A and ATP10D, with ATP10D being a faster transporter than ATP10A [44]. It was proposed that ATP10A and ATP10D, or their yeast orthologs Dnf1 and Dnf2, respectively, participate in a functional clade of glycosphingolipid flippases. Dnf1 and Dnf2 were both shown to be involved in the uptake of glucosylceramide and galacotsylceramide in yeast [44]. In addition, ATP10B has also been demonstrated to be a GlcCer transporter [44]. It is noted that the glycine-alanine (GA) motif in TM1 and the tyrosine-glutamine-serine (YQS) motif of TM4 were essential for the transport function. An amino acid substitution in the first or second position after the GA motif leads to a significant decrease in the transport of GlcCer. However, the recognition of PS and PE are not altered. Using substitutions, the study determined the most important helices for glucosylceramide transport: TM1, TM2, and TM6, where GlcCer clusters in the membrane rafts [45]. GlcCer metabolism disruption has been linked to lysosomal storage diseases such as Gaucher disease and Parkinson’s disease [46]. GlcCer also plays a role in multidrug-resistance of cancer cells [46,47].

Several residues on P4-ATPases are important in recognizing or transporting phospholipids. For instance, it was discovered that the glutamate located at position 215 was critical for the transport of glucosylceramide by ATP10D, whereas glutamate at position 203 was the most important for the transport of phosphatidylcholine by ATP10A [44]. The isoleucine found at position 364 of ATP8A2 is critical in neuronal development, where a point mutation at I364 on TM4 is associated with cerebellar ataxia, mental retardation and disequilibrium syndrome [36]. The vital role of isoleucine at position 364 leads to a hypothesis that the hydrophobic residues function as a hydrophobic gate to separate the entry and exit sites of the substrate [36].

### 2.3. Lipid Scavenging

Sometimes it is more beneficial for a cell to scavenge phospholipids from its environment rather than generating it from scratch [48]. Riekhof et al. [49,50] suggest that PE can be gathered by *S. cerevisiae* from nearby decaying organic matter by scavenging their lysophospholipid form via Dnf1p or Dnf2p in conjunction with their β subunit, Lem3p [49,50,51,52] (Figure 1D). In the case of PE specifically, the traditional method of generation would be through the Kennedy pathway [53]. A plant P4-ATPase, ALA10, has been proposed to have a role in scavenging soil lipids since it is located at the root tip (Figure 1D) [54]. Future studies could include looking into the roles of human P4-ATPases in lipid scavenging. Although *S. cerevisiae* has lost its ability to synthesize GlcCer, its P4-ATPases may have remained in order to scavenge the lipid from its environment [55,56].

### 2.4. P4-ATPases and Cellular Signaling

Lipid concentration can affect the activity of P4-ATPases. The Graham laboratory has shown that Dnf1 and Dnf2 can be regulated by ergosterol, and sphingolipids in yeast [57]. The PC flippase Dnf2 is autoinhibited and phosphorylation of Dnf2 removes the inhibition [39,40,57]. P4-ATPases are downstream targets of Kes1p, an oxysterol binding protein. Elimination of Kes1p has been associated with increased activity in the TGN, resulting in a greater number of protein-transporting vesicles. The P4-ATPase antagonizes Kes1p, so in *drs2*Δ cells, ergosterol transport from the PM to the ER was greatly increased [58] (Figure 1D). Drs2 is essential for segregating cargo into exocytic vesicles correctly [59]. In *C. elegans*, the P4-ATPase TAT-1, closely related to yeast Drs2, had a fine-tuning role in sterol metabolism [60] TAT-1 has also been associated with the formation of lysosomes [61] and ectosomes [62].

Interestingly, inhibition of sphingolipid biosynthesis reduces GlcCer transport by P4-ATPases in humans [57]. Sphingolipid metabolic pathway dysregulation is associated with pathologies such as cancer, cardiovascular disease, and type II diabetes [63]. For instance, ATP10A may serve a role in increasing GlcCer available which inhibits the insulin signaling pathway [64]. PS flippases, on the other hand, require phosphatidylinositol-4-phosphate (PI4P) binding to reactivate the flippase from autoinhibition [39,57]. PS exposure leads to membrane bending, promoting the creation of microvesicles, a process that inhibits extracellular vesicle (EV) release (Figure 1D) [65] [63,66,67]. Collectively, this suggests certain P4-ATPse-mediated mechanism that regulates the mechanic nature of cell membranes and is involved in EV release (Figure 1D). ATP9A, for example, serves as important regulator in several processes such as blood clotting, immune responses, and angiogenesis [66]. In cells where ATP9A was either overexpressed or underexpressed, sphingolipid metabolism was disrupted [63]. ATP9A depletion resulted in changes in cell proliferation, cell death, and survival pathways. This led to PS exposure (an apoptotic signal), recruitment of scramblases [68], inhibition of exosome release [66], and PS clustering at the location of lysosome-PM fusion [69]. PS exposure has been shown to be important for phagocytic recognition, which is important in neuron pruning [70]. In addition, mutations in P4-ATPases have been associated with neurodegenerative pathologies in which membrane asymmetry is lost [70]. For instance, P4-ATPases are implicated in myotube formation (Figure 1D). Specifically, ATP11A increases PS which activates PIEZO1, a molecule that governs morphogenesis during myotube formation [71].

P4-ATPases are regulated by kinases in many mammalian cells [72,73]. Internalization of ATP11C is induced by Ca^2+^-dependent PKCα activation and is mediated by clathrin-dependent endocytosis (Figure 1D) [68]. When Ser1116, found in the C-terminal cytoplasmic region of ATP11C, is phosphorylated, it generates a function di-leucine motif required for endocytosis [68]. Fpk1p/2p are upstream activators of Lem3p-Dnf1p/Dnf2p [72]. In addition, ATP11C may be important for bile acid transporter regulation since ATP11C deficiency leads to hyperbilirubinemia and hypercholanemia in mice [68,74]. Platre et al. [75] discovered that the plasma membrane’s electrostatic territory, which is defined as the combination of negatively charged anionic phospholipids, can localize cellular factors along the endocytic pathway. The electrostatic gradient was thus correlated with the presence of P4-ATPases [75].

### 2.5. Membrane Curvature

It has been postulated that the membrane curvature and the spatiotemporal lipid flippase activity can mutually regulate each other in recent studies [76]. Using BIN/amphiphysin/Rvs (BAR), protein domains that bind preferentially to curved membranes, curved membranes were detected cells after transfecting exogenous ATP10A in the HeLa, where no endogenous ATP10A is expressed [76]. This displayed a direct evidence that phospholipid-flippases can regulate the membrane curvature, and therefore cell morphology [76]. The availability of novel methods like this provides toolkits to track not only membrane curvature, but also flippase activity. Further studies will be key to determine the molecular mechanism of P4-ATPase in membrane morphology and cell signaling. In addition, CDC50 is important in the endocytic recycling pathway and specifically involved in the formation of vesicles from early endosomes [77,78]. ATP10A, found to decrease cell adhesion and spreading increased PC ration between the cytoplasmic and luminal leaflets, which would favor a positive curvature (Figure 1D) [79].

### 2.6. Coat Protein Recruitment

One method of endocytosis and the formation of endosomes is through clathrin-mediated endocytosis [80]. In 2003, Hinners and Tooze [81] demonstrated that clathrin-mediated transport can occur from the trans-Golgi network (TGN), which is a major secretory network, to endosomes, and vice versa [81]. The P4-ATPase Drs2p is an essential accessory protein for clathrin-mediated endocytosis (Figure 1D) [82]. Drs2p was required for the formation of exocytic vesicles and retrograde vesicles for the clathrin-mediated endocytic pathway [82]. Drs2p begins the vesicle formation through membrane curvature where the surface area of the cytosolic leaflet is increased in comparison to the luminal leaflet. This facilitates the capture and molding of coat proteins to produce vesicles [82]. Furthermore, it has been suggested that the negative nature of PS can produce a recruitment signal for coat proteins when present at a high enough concentration [83]. Chen et al. [84] showed that there were fewer clathrin-coated vesicles in *drs2*Δ than in WT [77].

### 2.7. Cytoskeleton Modulator

ATP9A regulates the actin network [66]. *ATP9A*Δ cells have been shown to have long, stabilized actin fibers [66,85]. ATP9A may also facilitate multivesicular late endosome (MVE) docking [63]. Defective ATP8A1 which remain on endosomes lead to aberrant fibrotic repair [78]. Actin nucleation and actin-based mobility were altered when ATP9A was depleted [66]. ATP9A depletion also affects cell morphology due to extensive cytoskeletal re-arrangements [66]. ATP10A increased the PC ratio between the cytosolic and luminal leaflets, leaving less room for PS and phosphatidylinositol 4,5-bisphosphate (PIP2) in the cytosolic leaflet. PS and PIP2 are both critical for actin cytoskeleton remodeling which occurs during cell adhesion, spreading, and migration (Figure 1D) [79].

## 3. ABC Sterol Transporters

### 3.1. ABC Transporter Structure

The ATP binding cassette (ABC) genes represent the largest superfamilies of transmembrane proteins in prokaryotic and eukaryotic organisms [93]. Several members of the ABC superfamily utilize ATP to actively mediate the transport of various substrates from the cytosolic leaflet outwards to the exoplasmic leaflet, thus also known as floppases [93,94]. A typical ABC transporter consists of four domains, including two nucleotide-binding domains (NBDs) and two transmembrane domains (TMDs), which together form the minimum functional requirement for molecule transport. The primary sequence of NBDs is evolutionary conserved, which defines the ABC superfamily. NBDs use the energy from ATP hydrolysis to power solute transportation across the lipid-bilayered membranes. The primary sequences of TMD can however differ across the ABC family. They generally form the translocation pathway across the cell membrane and can determine substrate specificity. Additional domains can be present in other ABC family members and may play a role as regulatory molecules [95]. To date, there are 44 known human ABC transporters categorized into five distinct ABC gene subfamilies (A, B, C, D and G) [96]. Many of these transporter proteins (ABCA, ABCB and ABCG in particular) are involved in lipid metabolism and are themselves regulated by lipids, including cholesterol, sphingolipids, phospholipids, and sterols [97]. Here, we will focus on the role of ABCA and ABCG transporters in cell membrane morphology and cellular signaling.

### 3.2. ABC Transporters in the Lipid Raft

Several ABC transport proteins are involved in the transport of bile, containing bile salts, organic ions, phospholipids, and cholesterol across the canalicular plasma membrane [98]. Ismair et al. [99] showed multiple, distinct phospholipid and cholesterol-enriched lipid microdomains (or lipid rafts) present in canalicular membrane in rat hepatocytes. Many ABC transporters reside within these microdomains through colocalization with caveolin-1, reggie-1, and reggie-2 microdomain markers [99]. It was proposed that a complex membrane lipid environment is required for the proper functioning of most ABC transporters. Moreover, cholesterol appears to be essential in the transport activity of ABCB1 and other ABC transporters by structuring the cell membrane and organizing the lipid rafts [100]. These findings demonstrated the importance of regulating cholesterol content in maintaining lipid rafts in the cell membranes [101,102,103].

Rafts are free-floating lipid microdomains contained within the plasma membranes. Exhibiting resistance to non-ionic detergents and existing in a liquid ordered (Lo) phase [104], they are enriched by sphingolipids and sterols. SM, having a high affinity for cholesterol, can form hydrogen bonds with cholesterol, subsequently leading to the generation and stabilization of raft structures [105,106,107]. In contrast to the Lo phase, the liquid disordered (Ld) phase of the surrounding non-raft regions features a loosely packed and detergent soluble membrane [108]. In addition to the regular raft components, the general term of lipid microdomains engulfs caveolae, characterized by the presence of cavin and caveolin proteins. This results in curved membrane invaginations roughly 60–80nm in diameter [109]. Varying sizes and the ability to cluster have given these elusive structures the option to form larger continuous domains throughout the membrane, a particularly important feature in cell signaling processes where non-specific interactions are permitted due to the proximity caused by such aggregating events [110,111] Thus, proteins and signaling pathways can be altered through such rafts.

### 3.3. Role of ABCA1 in the Intramembranary Cholesterol Movement

ABCA1 is a lipid transporter and is believed to function favorably in the SM-decreased Ld phase of the plasma membrane, whereas the Lo phase of the membranes might be required for ABCG1 function. [112,113]. It actively moves cholesterol and phospholipids from the cytosolic leaflet towards the extracellular leaflet while being picky with its lipid substrates, having a higher inclination for PC over SM [114]. Possessing the ability to translocate PS outwardly into the outer leaflet, ABCA1 has been linked to the phagocytosis of apoptotic cells [115]. Mutations in ABCA1 can cause Tangier’s disease, a genetic disorder marked by low HDL and ApoA1 levels causing an increased risk of atherosclerosis and peripheral neuropathy [116,117]. On the plasma membrane, ABCA1 regulates cholesterol redistribution from raft towards non-raft domains by promoting cholesterol efflux towards ApoA-1 (Figure 2C) [108]. In vitro, functional ABCA1 expressing cells had 50% more cholesterol available for extraction by cold Methyl-ß-cyclodextrin (MßCD) [112,118]. The mechanism in direct transport of cholesterol remains to be determined [108,114].

Using yellow fluorescent protein (YFP) linked caveolin as a marker for rafts to study the effects of ABC expression on raft formation, observations through confocal microscopy found that YFP-Caveolin was aggregated and localized in the non-functional ABCA1 and control cell lines [108]. Inversely, ABCA1 expressing cells had fluorescent caveolin more dispersed across the plasma membrane. Subsequent studies on ABCA1-expressing cells showed more detergent soluble membrane (DSM) [105]. These results should be interpreted with cautions. For instance, factors such as temperature and detergent concentration have been shown to affect protein organization and composition of rafts [105,119]. Limited information of the detergent resistant membranes (DRM) may hinder our interpretation about the raft composition [105,120]. New methods of more direct detection of cholesterol content [121] would provide direct evidence on ABCA1-mediated raft formation, subsequently shedding light on more defined model of rafts and the lipid composition.

### 3.4. ABCG Cholesterol Transporters Sinking Rafts

Human ABCG transporters function primarily as sterol transport [96]. ABCG1, similarly to ABCA1, mediates the transport of cholesterol and phospholipids with a liking for SM over PC, but catalyzes lipid efflux towards mature HDL [94,122,123]. Lowered SM membrane levels have been shown to antagonize ABCG1 function while upregulating ABCA1, with ABCG4 remaining unaffected, probably due to distinct local membrane environments [112]. Located in distinct DRM regions, ABCG1 and ABCG4 are thought to be contained within rafts. Evidence supporting this theory comes from the co-localization of ABCG1 with flottilin-1 (FLOT-1) and the partial co-localization of ABCG4 with the same raft marker [112]. Biochemical assays also show ABCG1 is fractioned in Triton X-100 rafts, while ABCG4 is soluble by Triton X-100 but found in Brij 96 rafts [112]. Interestingly, ABCA1, ABCG1 and ABCG4 have been shown to cooperatively function in removal of excess cholesterol acting successively to accomplish the task [112,124]. Similarly to ABCA1, these two ABCG transporters disrupt rafts formation through redistribution of cholesterol from raft to the surrounding membrane (Figure 2C) [108,112,123].

It has been known that ABCG2 directly interact with the caveolin-1 (cav-1) protein [125], but the physical interaction between ABCG1 and cav-1 is unclear and remains controversial. To understand how ABC sterol transporters regulate membrane microdomains, Sano et al. [112] further explored the ability of ABCG1 to reshuffle caveolin-1 protein from caveolae to non-raft regions. On one hand, inhibition of cav-1 binding to ABCG1 resulted in lowered cholesterol efflux [126], where the opposite was observed during interaction between cav-1 and ABCA1. When bound to the transporter, endocytosis activity was increased with the subsequent destruction of cav-1 and transporters [127]. In the case of ABCA1, binding of cav-1 acts as a way to prevent the disturbance of caveolae domains. Therefore, while the mechanism behind the eviction of cav-1 from caveolae domains remains unspecified, it is after all possible to hypothesize that a cascade of lipid interactions leads to an eventual protein interaction. The next step would be to figure out how ABC transporters interact with caveolin-1. Understanding the mechanism may reveal other proteins affected by the cascade, essentially showing further ramification caused by these transporters. The mystery of direct in vivo evidence does however linger, such evidence would support or reject the current raft hypothesis and guide future research in this field.

### 3.5. Twin Brothers in Apoptosis?

Studies of atherosclerotic foam cells have taken an interesting turn associating ABCG1 and ABCG4 with apoptosis, where a sterol-enriched environment is correlated with high ABCG1 expression [128,129,130]. Using annexin V (Axn-V) and caspase 3 (casp-3) activation as markers of apoptosis [131,132], functional ABCG1 cells had more PS translocation to the outer leaflet and higher casp-3 activation, as monitored by Axn-V binding assays. The deactivating ABCG1 mutant did not show such activity, suggesting the essential role of ABCG1 in apoptosis. These studies led to speculation of similar cellular role of ABC lipid transporters, including ABCG4.

Since ABCG4 and ABCG1 are so similar in terms of genetic markups and sequence homology, it has been believed that they are tied to apoptosis together [130]. Recent studies by Hegyi et al. [129] have shown that ABCG4 was associated with apoptosis [131]. WT ABCG4 not only stimulated higher Axn-V binding, but also higher casp3 activation. Interestingly, ABCG4-activated apoptosis is weaker when compared with that by ABCG1, among whose isoforms, the more frequently found short form (ABCG1S) has more profound effect on apoptosis [129,133]. Taken together, while the exact mechanism remains to be determined, ABCG cholesterol transporters clearly contribute to cellular signaling in apoptotic regulation.

### 3.6. Sterol-Sensing Domain (SSD) Tweaking Protein-Lipid Interactions

Allosteric binding sites are common in proteins, allowing substrates to bind in areas other than the active binding site to produce changes in the overall protein conformation. Exemplified by sterol sensing in proteins like Niemann Pick C1 (NPC1), cholesterol binding allosterically affects the protein morphology and then regulates the protein function [134]. SSD is believed to bind accessible cholesterol as their substrates, this being a minority group of sterols found in the membrane representing an unsequestered, unbound and active version of cholesterol [135,136]. In the hedgehog (Hh) signaling pathway, accessible cholesterol induces signal transduction through covalent bonding to the Sonic Hedgehog protein (SHH), a process resulting in the formation of an inhibitor to the Patched 1 (PTCH-1) protein. PTCH-1 itself is an inhibitor to Smoothened (SMO), a protein that transmits the Hh signal through the membrane from the exoplasm into the cytoplasm. This communication is governed by oxysterol or cholesterol giving them the role as secondary messengers [137].

Similarly, cholesterol and SM bound to ABCG1 can stimulate ATPase function and cholesterol affinity respectively [122], suggesting multiple cholesterol-binding sites on ABCG1. When investigating an ABCG1 homology model built based on the crystal structure of ABCG5/G8, transmembrane regions reveal the locations of the putative SSD containing a conserved cholesterol recognition/interaction amino acid consensus motif (CRAC) (Figure 2A,B). In a recent study, these cholesterol recognition regions were found to be important in the proteins stability with Y667 in particular deemed essential [135]. The Y667L mutant led to a complete loss of cholesterol sensing ability, whereas the other mutants, Y649L and Y672L, showed decreased cholesterol efflux but were still able to be stabilized by added cholesterol. As seen in Figure 2B, when aligning the sequences of various ABCG proteins, the high conservation of Y667 is hard to go unnoticed; all but ABCG5 contained this tyrosine. As ABCG5 and ABCG8 function as an obligatory heterodimer, one subunit may be sufficient to this SSD. The necessity of Y667 is thus deemed to proper function of ABCG cholesterol transporters.

Membrane cholesterol thus appears to regulate its own plasma content through the interaction with CRAC motifs on cholesterol-binding proteins, and in the case of ABCG1, such content change of cholesterol is coupled to ABCG1′s ATPase activities. As ABCG1 pertains the ability to redistribute cholesterol against the concentration gradient, it is possible that when the accessible cholesterol concentration reaches a threshold and binds, ABCG1 is upregulated with intent to restore baseline levels and maintain cholesterol homeostasis within the plasma membranes. This feature would then allow ABCG1 to sense and interfere the raft domains as mentioned previously.

### 3.7. ABC Transporters and Cellular Signaling

Plasma membranes not only serve as a structural barrier but also an integral source of lipids, participating in various signal transduction pathways. The role of lipids as ligands can be achieved through activating signal transduction, mediating signaling pathways, and as substrates for lipid kinases and phosphatases [138]. Continuous improvement in analytical methods makes it possible to study the position and dynamics of individual molecules, such as proteins and lipids, in real-time. Live-cell single-molecule fluorescence imaging has become a powerful analytical tool to investigate such cellular processes [139]. Recent advances in single-molecule imaging were developed, which greatly enhanced the imaging time resolutions down to 5.0 ms, an improvement by a factor of 6.7 [140]. The resolution substantially refined the detection of signal transduction mediated by CD59 cluster rafts and raftophilic lipid rafts in the outer leaflet of the PM. Dependence on the lateral raft–lipid interactions with cholesterol and molecules with saturated alkyl chains can now be measured by following the stabilized raft-lipid clusters upon the recruitment and activation of cytoplasmic signaling molecules, such as Lyn, H-Ras, and ERK, at the inner leaflet [140]. The result suggests a direct involvement of cholesterol in signal transduction. However, involvement of transmembrane proteins has not been characterized using this approach yet, which deems new findings on transporter recruitment to enhance signaling at the raft domains.

The mechanism of how ABC transporters connect signal transduction and membrane cholesterol content however remains to be defined. A recent study showed that some effects of ABC transporters on signal transduction might be imparted by their influence on cholesterol transport for raft formation (Figure 2C). One example is that reduction of ABCA1 through in vivo electroporation with shABCA1-RFP plasmid in mice resulted in cholesterol efflux defect and the subsequent increase in the membrane cholesterol content [141]. Increased membrane cholesterol reduces phosphoinositide-dependent kinase (or Akt) phosphorylation at Ser-473, insulin-induced GLUT4 translocation in the plasma membrane, and glucose uptake in skeletal muscles (Figure 2C) [141]. Similarly, an abnormal status of ABCA1 is observed in insulin resistance [142]. In contrast, increased ABCA1 and ABCG1 content in polydatin-treated microglia cells significantly inhibits lipid rafts formation through membrane cholesterol depletion. The disrupted lipid rafts prevent the phosphorylation of phosphoinositide 3-kinase (PI3K) and Akt; thus, blocking the downstream nuclear factor-κB (NF-κB) signaling pathway (Figure 2C) [143]. NF-κB is a transcription factor involved in multiple aspects of the innate and adaptive immune systems, such as regulating immune response through stimulating pro-inflammatory genes [144]. Unregulated NF-κB is associated with many inflammatory diseases [145]. On the other hand, activated Akt can phosphorylate as many as hundreds of downstream cellular substrates, leading to inhibition or activation of various cellular pathways involved in cell growth, death, and survival [146]. Thus, ABC transporters regulate membrane cholesterol content as a consequence of cholesterol efflux, ultimately affecting signal transduction, such as insulin-dependent signaling pathways, NF-κB signaling pathways, and PI3K/Akt phosphorylation (Figure 2C).

In the central nervous system (CNS), ABCA1 plays a critical role in the lipidation of ApoE and HDL cholesterol and lipid metabolism [147,148]. Like other biological membranes, myelin in the CNS is abundant with phospholipids and cholesterol content, creating an environment compatible with the existence of lipid rafts [149,150]. A neutral lipid-enriched high-fat diet promote myelination along with elevated levels of key cholesterol transport proteins, including ApoE and ABCA1 [151]. However, the mechanism underlying the ABCA1/ApoE interaction in myelin repair is not well understood. Recently, a proof-of-concept underlying the ABCA1/ApoE/HDL pathway mediates myelination during stroke repair [152] ABCA1 knocked down in stroke mice showed a significant decreased in myelinated axons and myelin sheath thickness. And intracerebral administration of ApoE and HDL in the knocked down ABCA1 stroke mice remarkably improved axonal myelination [152,153]. Although the findings did not characterize downstream signaling molecules, ABCA1/ApoE/HDL signaling pathway may contribute to the trafficking and sorting of lipids localized in lipid rafts for proper myelination.

One interesting perspective of ABC cholesterol transporter-mediated signal transduction came from viral pathogens. Intracellular pathogens, such as the human immunodeficiency virus (HIV), target lipid rafts of the host PM and play a role in cholesterol metabolism to alter cellular functions and cause inflammation [154]. HIV protein Nef can inhibit cholesterol efflux of ABCA1 by displacing ABCA1 from the lipid rafts with its subsequent degradation [155]. Consequently, reduced ABCA1 leads to increased lipid rafts formation through impaired activation of small G-protein Cdc42 followed by a reorganization of the actin cytoskeleton and altered downstream kinases (PAK-1 and p54JNK) [156,157,158]. The increase in membrane lipid rafts caused TREM-1 and TLR4 to be recruited into the raft, leading to TLR4 activation, ERK1/2 phosphorylation, inflammasome activation, and pro-inflammatory cytokines secretion (Figure 2C) [157]. Treatment with Selenium on lipopolysaccharide (LPS)-induced inflammation in mice activate the nuclear receptor liver-X-receptor (LXR)-ABCA1 pathway, promoting cholesterol efflux and inhibiting raft formation [159]. The activation of LXR also induces the expression of other ABC genes, including ABCG1, ABCG5, and ABCG8 [160]. The effect is the impediment of TLR4 migration to the lipid rafts and to prevent the inflammatory response caused by LPS (Figure 2C) [159]. Thus, ABC cholesterol transporters are involved in mediating inflammatory response through regulating rafts interaction with actin cytoskeleton and other transmembrane proteins, such as TREM-1 and TLR4. Collectively, recent evidence supports the functions of ABC transporters in signal transductions through regulating cholesterol content in the lipid rafts.

## 4. Conclusions and Perspectives

(i) Lipid transporters and membranes working in consort: Individual membrane lipids are known to regulate cellular signaling, such as phosphatidyl inositides or diacy-glycerides [161,162]. As highlighted above, protein functions, especially membrane proteins, can be regulated by changes of lipid composition and/or microdomains in the cellular membranes. The dominant lipid components of plasma membranes are sterols, phospholipids, and sphingolipids. In addition to passive permeation and communication barriers, emerging evidence has shown that changes in the membrane environment or morphology can regulate signaling functions in the cells. Such process is dynamically coupled to membrane protein functions on the plasma membranes. Studied as separate entities, on one hand, phospholipids interact with cholesterol in model membranes; on the other hand, sterols or certain species of phospholipids can regulate the activities of membrane proteins. In recent years, more and more advancement of experimental techniques and biophysical/biochemical methods allows us to investigate membrane proteins and their surrounding membrane environment together, a new frontier in membrane biology and cell signaling. Lipid transporters in particular play an important role in maintaining the lipid pools of the membranes. Mechanistic questions to address include: “how the lipid transporters recognize the needs in the membranes and subsequently activate or repress their functions?”, “what types of lipid environment on the membrane will be sufficient to trigger or recruit the intracellular signaling events?”, or “how the interaction among the transporters, the membranes and the intracellular messengers regulate each other?” As highlighted here, the phospholipid-transport P4-ATPases and the ABC cholesterol transporters not only regulate lipid asymmetry by an ATP-dependent lipid translocation in the membranes, but also are responsible for physical changes of membrane shapes in response to external or internal signaling events of cells. Both the lipid transporters and the membranes, while seemly separate entities, have to work together to maintain their functions, as well as transmit signals across the lipid-rich membrane barriers.

(ii) Crosstalk between different transporter types: Members of P4-ATPase or ABC lipid transporters can be found overlapped in multiple tissues (Human Protein Atlas: http://www.proteinatlas.org, accessed on 22 July 2021) [163,164,165]. Both types of lipid transporters share the common natural environment to function, lipid bilayered membranes. Phospholipids have been shown to regulate the functions of ABC cholesterol transporters, and sterols has been shown to regulate P4-ATPase activities in yeast. Both phospholipids and sterols are the physiological substrates of P4-ATPase and ABC lipid transporters. It will come as no surprise to learn that P4-ATPase and ABC transporters are regulated by common intermediates, such as changes of membrane environment or shared signal transduction pathways. The crosstalk between different types of transporters unfortunately remains understudied. With the advancement in membrane protein reconstitution [166,167], protein structural analysis of cellular events at near atomic resolution by cryo-electron tomography [168], and biophysical methodology of multi-subunit macromolecules, the existing data of cell biology, biochemistry, and structural biology on P4-ATPases and ABC lipid transporters should promote interdisciplinary biochemical, biophysical and computational approaches to elucidate the cross-talk among different lipid transporters in regulating membranes and membrane-mediated cellular signaling.

## Figures and Tables

**Figure 1 membranes-11-00562-f001:**
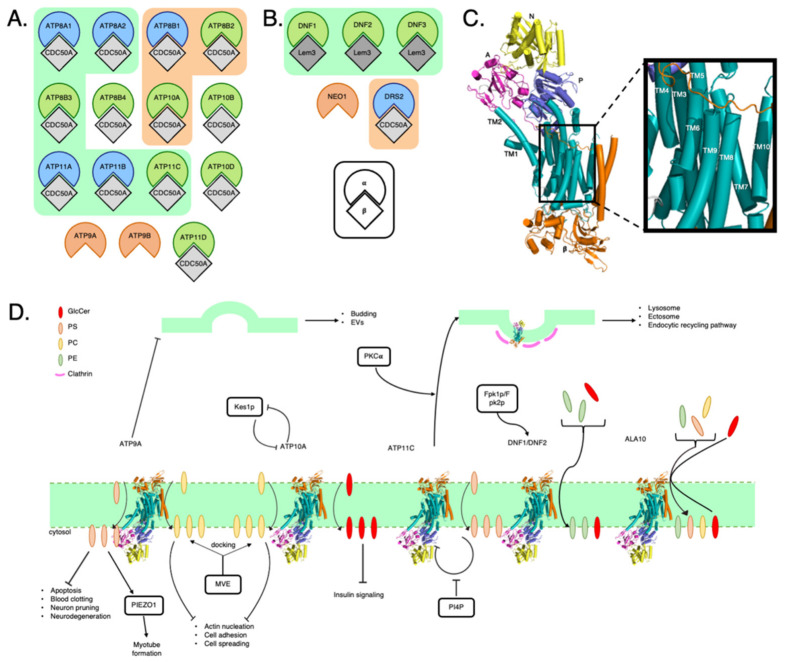
(**A**) Fourteen human P4-ATPases are shown, each corresponding to a different homologous group. The α subunit is bound to the β subunit. The five P4-ATPases found on the green background all have PS or PE as a transport substrate. All three P4-ATPases on the orange background have PC as a substrate. Three P4-ATPases do not have a beta subunit. The 14 human P4-ATPases paired with their respective beta subunit [77,84,86,87,88,89,90,91] (**B**) Five *S. cerevisiae* P4-ATPases. Each P4-ATPase is paired with its respective beta subunit. The P4-ATPases on the green background all have PS or PE as a transport substrate. The P4-ATPases on the orange background have PC as a substrate [77,84,86,87,88,89,90,91]. (**C**) A structural model of P4-ATPase. The cytoplasmic domains are shown above. The nucleotide binding (N) is shown in yellow, the phosphorylation (P) is shown in blue, and the actuator (A) domain is shown in magenta. The 10 transmembrane helices (TM1-TM10) are shown in teal. The β subunit is shown in orange [42]. (**D**) Cellular signaling associated with P4-ATPases. ATP9A is a human P4-ATPase which flips phosphosphingolipids (PS) which has been shown to inhibit apoptosis, blood clotting, neuron pruning, and neurodegeneration [66,68,70]. The accumulation of PS on the cytosolic leaflet has also been shown to activate PIEZO1 downstream which governs morphogenesis during myotube formation [71]. PS accumulation on the luminal leaflet has been linked to budding in budding yeast and extracellular vesicles (EVs) [66,92]. ATP9A and ATP10A flip phosphatidylcholine (PC). The increased PC on the cytosolic leaflet has been shown to be a docking site for multivesicular late endosomes (MVE) [63]. High cytosolic leaflet PC concentration has been shown to inhibit actin nucleation, cell adhesion, and cell spreading [63,79]. ATP10A transports glucosylceramide (GlcCer) to the cytosolic leaflet which has been shown to inhibit insulin signaling [64]. Kes1p is an upstream oxysterol binding protein which targets P4-ATPases which antagonize Kes1p in return [58]. PS flippases such as ATP11C are autoinhibited, and phosphatidylinositol-4-phosphate (PI4P) inhibits the autoinhibition, i.e., reactivates the flippase [39,57]. PKCα activates ATP11C which leads to the internalization of the flippase through clathrin-mediated endocytosis [68,77,80,82,83]. P4-ATPases have also been implicated in the formation of lysosomes, ectosomes, and the endocytic recycling pathway [61,62,69].

**Figure 2 membranes-11-00562-f002:**
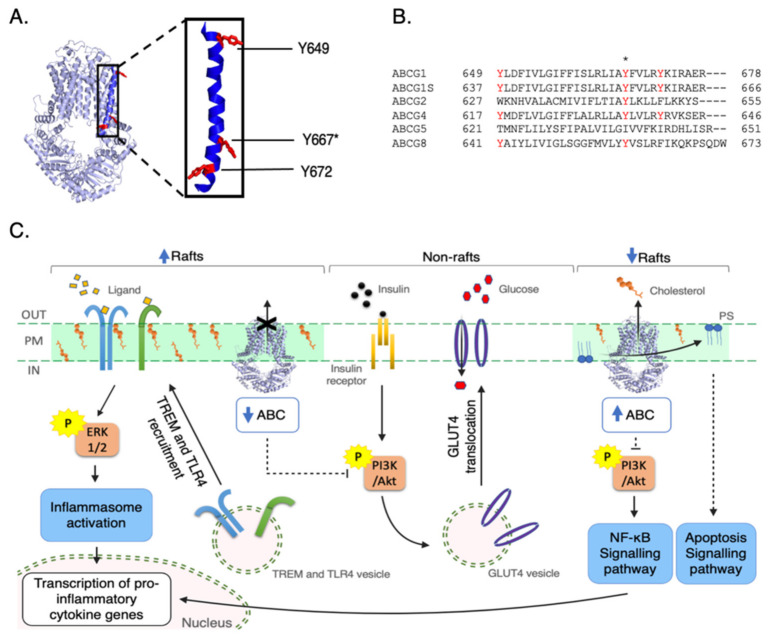
(**A**) Homology model of the human ABCG1 homodimer. Using the human ABCG5/G8 crystal structure (PDB ID: 5DO7) as a template, the sequence alignment was first generated by PROMOLS3D, using ABCG5 for one subunit and ABCG8 for the other. Calculated with the program MODELLER, the best model was selected from 100 results based on the best score of the discrete optimized protein energy (DOPE). Three highlighted tyrosine’s located on the same alpha helix representing the proposed CRAC (Cholesterol Recognition/interaction Amino acid Consensus) motifs labelled Y649, Y667 and Y672 accordingly. Solely Y667 is marked with an asterisk (*) due to its essential nature in the overall function and stability of ABCG1 [135]. (**B**) Sequence alignment displaying conservation of CRAC motif in various ABCG family transporters. Done using PROMALS3D, highlighting (red) the conservation of CRAC motifs in human ABCG family transporters (ABCG1, ABCG1S, ABCG2, ABCG4, ABCG5, ABCG8). Motif 667 (*) showing very high conservation throughout all but ABCG5, an obligatory heterodimer with ABCG8, a CRAC motif containing protein. (**C**) ABC transporters in signal transduction through regulating plasma membrane (PM) cholesterol content in the lipid rafts. Increase in ABC transporters (i.e., ABCA1, ABCG1, ABCG4, ABCG5/8) promote cholesterol efflux, thus decreasing cholesterol content within the PM and reducing raft formations [143,159]. This inhibits phosphorylation of phosphoinositide 3-kinase (PI3K) and phosphoinositide-dependent kinase (Akt), blocking the downstream Nuclear Factor-κB (NF-κB) signaling pathway, and there-by impeding TLR4 migration to the lipid rafts, and ultimately preventing the inflammatory response [143,159]. Additionally, phosphatidylserine (PS) is translocated through ABCG1 from the inner to the outer leaflet of the PM serving as an apoptotic marker and resulting in cell death [115]. Conversely, a decrease or defect in ABC transporters may lead to membrane cholesterol accumulation, increasing lipid rafts formations [141,155,156,157,158]. This recruits TREM-1 and TLR4 into the raft, leading to their activation, ERK1/2 phosphorylation, inflammasome activation, and pro-inflammatory cytokines secretion [157]. Increased lipid rafts further reduce PI3K/Akt phosphorylation, insulin-induced GLUT4 translocation in the PM, and glucose uptake in skeletal muscles [141]. Direct pathway (solid arrows); indirect pathway (dashed arrows).

## Data Availability

Not applicable.

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
