# Peer review of "Lipid Transporters Beam Signals from Cell Membranes"

_membranes, 2021, doi:10.3390/membranes11080562_

Round 1
Reviewer 1 Report
Ristovski et al. "Transporters beam signals from cell membranes"
The authors make an excellent review of the knowledge and recent advances on the field of the involvement of lipids such as phospholipids and sterols, and their corresponding transporters at the plasma membrane.
There is but one "serious" mistake: Figure 2 is an orphan. While figure 1 is appropriately called from the text, Figure 2 has no main text callings. A pity because it is a very informative figure.
Just a few minor mistakes that need addressing:
Page 3, line 113: it should read "[...] regulated by ergosterol, and sphingolipids"
Page 3, line 123: it should read "[...],so in drs2Δ cells, ergosterol transport [...]". The yeast consensus dictates that gene three-letter names are written in capital italics when wild-type and all small case italics when mutated.
Reviewer 2 Report
In the manuscript entitled “Transporters beam signals from cell membranes” the authors present a literature revision regarding the function of the lipid transporters P4-ATPase and some ABC transporters, focused on cell signalling. Several recent references are cited, and an attempt is done to interpret the information from the work cited. The manuscript may therefore be of utility to researchers working in the field. There are however several issues that must be corrected, mostly to improve clarity and simplify reading. Some specific concerns are given below:
1 – The title is confusing at first, consider replacing “beam” with “send”, or maybe give a more specific and informative title.
2 – There are some grammatical incorrections and a few typos throw-out the text. Just as a few examples: verbal tense in line 27/28; incomplete sentence and/or singular/plural inconsistency in line 58; lines 507-509 “we have learned… we gained… we have”, scientific advancement is not about “we” the knowledge should be evident for everyone; lines 517/518;
3 – Some information is repeated too many times, making the manuscript unnecessarily long. Just as a few examples: Line 62 “Each monomeric P4-ATPase consists of ten transmembrane domains …” and on line 65 “P4-ATPases have 10 transmembrane helices: TM1 to TM10 [42]”; line 160/163, “Using BIN/amphiphysin/Rvs (BAR), protein domains that sense membrane curvature by binding preferentially to curved membranes, curved membranes could be detected by the BAR domains in HeLa cells where no endogenous ATP10A is expressed [77].”
4 – Some information is not properly contextualized and is very difficult to follow. Just as a few examples: section 2.1 – after the general discussion on P4-ATPases, specific reference to Drs2p or ATP8A without a proper introduction; lines 80/81 “…GA motif …. YQS motif”; line 103/104 “lysophylized form”?
5 – Sometimes the message is confusing or even inconsistent or incorrect. Just as a few examples: The authors go back and forth through the different human and S. cerevisiae P4-ATPases without a clear and/or simple order or organization; the substrate specificity given in the caption of Figure 1 is not always in agreement with the information in the text (eg lines 83/84); P4-ATPases in Figure 1 are shown with the cytoplasmic domains towards the extracellular media; lines 95-99, is asparagine an hydrophobic residue?; line 154, “electrostatic territory”, what is it?; line 164, what is “exogenous ATP10A”?; on section 3.1 it is said that focus will be given to ABCA and ABCG, but section 3.2 starts with ABCB1; section 3.3, is ABCA1 localized in the Ld phase? What are the differences (structural and functional) between ABCA1 and the other ABC transporters that localize in Lo domains?; section 3.4, the title indicates ABCG1, but the text includes many other ABC transporters; lines 305-307 “Evidently, the most important unsolved mystery is direct in vivo evidence as mentioned previously, making or breaking the current raft hypothesis and allowing a proven basis to be established and built upon.”; line 367 “time resolutions down to 5.0 nm”; 418 “…with cytoskeleton and other transmembrane proteins”; section 3 must be re-organized, all aspects indicated are a consequence of cholesterol efflux by the ABC transporters (or lack of it when they are down regulated) with the resulting decrease (increase) in membrane cholesterol levels.
6 – Some sentences should include references to support the information provided. As a few examples: lines 84/85, 93/94, 225-228,
7 – Too many unexplained abbreviations.
Round 2
Reviewer 2 Report
The authors have addressed most of the reviewer concerns.